# A Mathematical Model for the Combination of Power Ultrasound and High-Pressure Processing in the Inactivation of Inoculated *E. coli* in Orange Juice

**DOI:** 10.3390/foods13213463

**Published:** 2024-10-29

**Authors:** Óscar Rodríguez, Vibeke Orlien, Ashwitha Amin, Emiliano Salucci, Francesco Giannino, Elena Torrieri

**Affiliations:** 1IRIS Technology Solutions, S.L. Crta d’Esplugues 39-41, 08940 Cornellà de Llobregat, Spain; orodriguez@iris-eng.com; 2Department of Food Science, Faculty of Science, University of Copenhagen, Rolighedsvej 30, 1958 Frederiksberg, Denmark; vor@food.ku.dk (V.O.); ashwitha@food.ku.dk (A.A.); 3Department of Chemical Engineering, Faculty of Science and Engineering, Åbo Akademi University, Henrikinkatu 2, 20500 Turku, Finland; emiliano.salucci@abo.fi; 4Department of Agricultural Sciences, University of Naples, Via Università 100, 80055 Portici, Italy; giannino@unina.it

**Keywords:** non-thermal technologies, power ultrasound, high-pressure processing, orange juice, mathematical model, system dynamics, ordinary differential equation, numerical simulations

## Abstract

The mathematical modeling of a combination of non-thermal technologies for *E. coli* inactivation is of great interest for describing the dynamic behavior of microorganisms in food, with the goal of process control, optimization, and prediction. This research focused on the design and implementation of a mathematical model to predict the effect of power ultrasound (US), high-pressure processing (HPP), and the combination of both non-thermal technologies on the inactivation kinetics of *E. coli* (DSM682) inoculated in orange juice. Samples were processed by US, HPP, and a combination of both technologies at varying process parameters, and a mathematical model for microbial inactivation was developed using a System Dynamics approach. The results showed that the combination of these technologies exhibited a synergistic effect, resulting in no detectable colony-forming units per mL of juice. The developed model accurately predicted the inactivation of *E. coli* following the combination of these technologies (R^2^ = 0.82) and can be used to predict microbial load reduction or optimize it based on process parameters. Additionally, combining both techniques offers a promising approach for extending the shelf life of fresh juices using non-thermal stabilization technology.

## 1. Introduction

The awareness amongst consumers when it comes to the consumption of fresh-like and healthy juices with desirable nutritional and sensory properties has increased [1]. Traditional stabilization methods like pasteurization and sterilization are widely applied by the juice industry due to the important role played by them in the extension of shelf life. However, it is well-known that traditional stabilization negatively affects the nutritional and sensory profile of the juices, which reduces the acceptance by consumers [2]. Non-thermal technologies could be considered as alternatives to traditional stabilization. They are defined as preservation treatments that are effective at ambient or sub-lethal temperatures, thereby minimizing negative thermal effects on food nutritional and quality parameters [3,4].

High-pressure processing (HPP) can be used as alternative stabilization process, able to destroy food-borne pathogens and inactivate enzymes while aroma compounds, pigments, vitamins, and other small-molecular-weight compounds are rarely affected. The main stabilization mechanisms involved in HPP are cell membrane destruction, protein denaturation, and DNA damage. The main factors affecting the stabilization effect of HPP are the pressure level, duration of treatment, temperature, pH level, food composition, microbial resistance, and food structure. Typically, industrial pressure levels for preserving fruit juices ranges between 400 and 600 MPa, with short holding times (maximum 5 min depending on the type of juice) [3]. However, the shelf life of HPP juices is still relatively short (compared to heat-treated/UHT-processed ones), and refrigeration conditions are critical for preserving the quality of the juices. Power ultrasound (US) has been reported to inactivate food enzymes and consequently to increase the shelf life of fruit/vegetable juices, while keeping their healthy properties [5,6,7]. The effect of US application on microorganism inactivation depends on external factors such as the acoustic power, frequency, and amplitude, and internal factors such as the juice matrix: its structure and composition [8,9]. The mechanisms involved in the weakening or destruction of the microorganism’s cell wall include the collapse of the cavitation bubbles within or around the microorganisms. The collapse causes a change in the pressure gradient, promoting cell wall damage. Although most microorganism can withstand high pressures, they are unable to withstand the quick alternating pressures produced during cavitation [8]. However, sonication alone is not effective for the decontamination of juice products, and its potency could be enhanced by a combination with other technologies [10,11,12,13].

The combined use of high-pressure processing (HPP) and ultrasound (US) significantly enhances microbial inactivation in food products. Increasing the amplitude of ultrasonic waves and applying higher pressure can improve the effectiveness of US [14]. When used together, HPP and US can exhibit both additive and synergistic effects, leading to greater reductions in pathogens [7,15,16,17]. This integration enhances food safety and preservation, making it a powerful approach in food processing.

Mathematical modeling of the HPP and ultrasound processes is an area of significant interest, as it enables the detailed analysis and understanding of the dynamic behavior of various species within food products, including microorganisms, enzymes, and other bioactive compounds. These models are crucial for process control, optimization, and prediction. In fact, mathematical models provide a framework for monitoring and controlling the conditions of HPP and US treatments. By simulating different scenarios, operators can identify optimal conditions that maximize microbial inactivation while preserving the quality of the food. Moreover, through modeling, it is possible to optimize parameters such as pressure, temperature, treatment time, and ultrasonic amplitude. This ensures that the processes are both effective and efficient, minimizing energy usage and maximizing product quality. Models can also predict the outcomes of various processing conditions on microbial inactivation and the stability of food components. This predictive capability is essential for designing processes that meet safety regulations and consumer expectations [18].

It has been demonstrated multiple times that when microorganisms are exposed to new non-thermal technologies, inactivation does not follow first-order kinetics. In this context, classical deterministic models based on first order kinetics are no longer applicable [18]. One stochastic alternative that has been recently used by different authors to describe nonlinear survival curves is based on the Weibull distribution function [19,20], whereas the Lorenzian model was applied to describe the nonlinear survival curves when thermosonication was used [19]. The inactivation of microbes and enzymes can be described by a number of different types of models, including log-linear, shoulder, and tail curves. These equations are dependent on a number of factors, including the process parameters, matrix characteristics, and the target of the inactivation. The occurrence of downward concave curves (i.e., an initial lag period followed by an inactivation phase) may be attributed to a number of biological factors, including sub-lethal injury, the repair of cellular damage, protein refolding, and the cumulative impact of damage exceeding a certain threshold. Conversely, upward concave (tails) curves may be attributed to the presence of subpopulations, wherein the more sensitive population is inactivated rapidly, allowing progressively more resistant survivors to emerge [21]. In this context, we designed a mathematical model of the microbial process using a System Dynamics modeling approach to capture the complexity of our system.

The System Dynamics approach is founded upon a formulation of the system in terms of ordinary differential equations (ODEs). The system is represented in graphical form using fundamental components: stocks, flows, variables, and influences [22]. This approach greatly simplifies the creation and communication of models and has impressive analytical capabilities [23].

Thus, in this research, the inactivation of *E. coli* inoculated in orange juice by US and HPP was carried out, and the main objective was to evaluate the inactivation kinetics of *E. coli* as a function of US and HPP process parameters. To this aim, both technologies were first studied independently and later simultaneously. Then, to better understand the synergistic effect of their combination, the specific objectives were (i) to model—by using a System Dynamics approach—the inactivation of *E. coli* as a function of US and HPP process parameters and (ii) to validate the mathematical model by predicting the inactivation effect of the combination of US and HPP.

## 2. Materials and Methods

### 2.1. Bacteria Stock Culture and Inoculum

Lyophilized *Escherichia coli* (DSM682, Leibniz Institute, DSMZ—German Collection of Microorganisms and Cell Cultures GmbH, 38124 Braunschweig, Germany) was added to Tryptone Soy Broth (TSB) (HIMEDIA (Thane, India); product no. M011) and grown for 24 h. An aliquot of 0.2 mL was added into 9.8 mL of TSB and then incubated for 24 h at 37 °C. Thereafter, 1 mL of culture was added into 9 mL of TSB, followed by incubation at 37 °C for 24 h. The final population of *E. coli* was defined by plating serial dilutions on agar media incubated at 37 °C for 24 h. Finally, 500 µL of the *E. coli* inoculum was added to 500 mL of commercial orange juice (Brähmults; pressed and pasteurized) in a beaker to obtain a final concentration between 5 and 6 log CFU mL^−1^. The inoculated juice was kept at room temperature for 1 h until the process [12].

### 2.2. Ultrasound Treatment

The power ultrasound application was conducted in batch mode using an UP400St (400 W, 24 kHz) ultrasonicator (Hielscher, Teltow, Germany). Sonication was carried out using different titanium probes with nominal amplitudes of 46 µm (probe A), 99 µm (probe B), and 164 µm (probe C). The acoustic power (HUS) delivered to the sample was calculated according to Margulis et al. [24] by using two different solutions: 90% glycerol (⍴_20 °C_: 1.235 gcm^−3^) and 60% sucrose (⍴_20 °C_: 1.288 gcm^−3^) [25]. The variation in mean temperature as a function of time was recorded at adiabatic conditions using a thermocouple (PT100), and the acoustic power was calculated according to Equation (1), where *M* is the mass of the sample (kg) and cp is the specific heat (J kg^−1^ K^−1^) of the sample.
(1)$$H_{US}=M\;\cdot \;C_{p}\cdot \frac{dT}{dt}$$

A half-liter of inoculated orange juice with an approximate initial population of 5 log CFU mL^−1^ was poured in a jacketed vessel (600 mL) with a cooling jacket connected to an CW-500 industrial chiller (S&A, Guangzhou, China) set at 5 °C to avoid the extra heating of the juice. The juice was sonicated in pulsed mode, 10 s ON 10 s OFF; at two amplitude levels: 50 and 100% during different exposure times from 10 to 60 min. An aliquot of 100 μL was taken at regular time intervals, while the rest of it continued being sonicated. The aliquot was kept under refrigeration until the microbial count by plating serial dilutions was done. 

### 2.3. HPP Process

The high-pressure process was carried out using QUINTUS Food Processing Cold Isostatic Press QFP-6 equipment (Avure Technologies AB, Västerås, Sweden). Three batches, A, B, and C, of inoculated orange juice were prepared in duplicates for the HPP and US + HPP processes: 500 µL of the *E. coli* inoculation solution was added to 500 mL of pressed and pasteurized orange juice (Brähmults) in a beaker, and then the mixture was kept at room temperature for 1 h. Thereafter, a small volume of each of the juice batches (A, B, or C) was placed in sterile Cryo.s tubes (Greiner Bio-One, Kremsmuster, Austria), and the tubes were then vacuum sealed in vacuum bags. The packed juice samples were high-pressure treated at 300 MPa for three different time points, i.e., 1, 5, and 10 min, and at 400 MPa for 1, 2, and 3 min. The pressure-transmitting medium used was 0.5% sodium benzoate water at 10 °C. Samples for pH and Brix analyses were frozen immediately after processing and stored until analysis, while samples for microbiology counts were analyzed right after the process.

### 2.4. Combination of Technologies: US + HPP Process

The combination of both technologies was carried out as follows: first, the US treatment of the inoculated orange juice (batches A, B, and C) was carried out using the UP400St ultrasonicator (400 W, 24 kHz) together with probe B (99 µm). Pulsed mode was implemented (10 s ON, 10 s OFF) to avoid excessive heating. The juice temperature was monitored during the treatment using a PT100 thermocouple inserted in the juice sample and kept below 22 °C by using a CW-500 industrial chiller (S&A, Guangzhou, China). The US application was extended for 720 s (batch A), 1440 s (batch B), and 2160 s (batch C). The acoustic power (H_US_) delivered by probe B (99 µm) was determined following the methodology described in chapter 2.2 prior to the experiments. After the US treatment, the ultrasound-treated juice samples were placed in sterile Cryo.s tubes (Greiner bio-one, Kremsmuster, Austria), and the tubes were then vacuum-sealed in vacuum bags. The packed juice samples were high-pressure treated at 300 MPa for 1, 5, and 10 min and at 400 MPa for 1, 2, and 3 min using a QUINTUS Food Processing Cold Isostatic Press QFP-6 (Avure Technologies AB, Västerås, Sweden). The pressure-transmitting medium used was 0.5% sodium benzoate water at 10 °C. US + HPP samples for pH and Brix analyses were frozen immediately after processing and stored until analysis.

### 2.5. Mathematical Model for Microbial Inactivation

The phenomenon of the microbial inactivation of microorganisms in new technologies is often difficult to interpret as it typically presents sigmoidal, nonlinear trends. The following model has been developed, modifying the mathematical model proposed by Geeraerd [26]. Ordinary differential Equations (2) and (3) describe the rate of inactivation of the microbial population (*N*) and the rate of formation of protective or critical components (*C_C_*), respectively:(2)$$\frac{dN}{dt}=-k_{i}\cdot N\cdot \left(\frac{1}{1+C_{C}}\right)\cdot \left(1-{\left(\frac{N_{i,min}}{N}\right)}^{n}\right)$$
(3)$$\frac{dC_{C}}{dt}=-k_{i}\cdot C_{C}$$
where *i* defines the type of technique used (US or HPP), *k_i_* represents the inactivation rate, and *N_i,min_* is the most resistant fraction of the population, while *n* is an exponential factor related to the tail effect of the treatment. In Equation (2), it is possible to observe the presence of two terms: a term describing the initial phase of the delay, i.e., the shoulder effect that may occur at the beginning due to the presence of a pool of *C_C_* components that must be inactivated before a significant effect on the target population can be achieved, and a term describing the tail effect due to a treatment-resistant subpopulation.

In addition to the shoulder and tail effects, the nonlinear trends may depend on many other aspects such as the chemical–physical properties of the product (pH and viscosity) or the operating conditions of the process (intensity). The *k_i_* and *N_i,min_* terms described in Equations (4) and (5) are defined to account for these aspects:(4)$$k_{i}=\frac{k_{i,max}}{\left(1+exp\left(-w_{i}\left(H_{i}-h_{i,c}\right)\right)\right)}\left(1+a{\left(pH-pH_{opt}\right)}^{2}\right)\left(1-\frac{{log}_{10}(\eta )}{{{log}_{10}(\eta }_{max})}\right)$$
(5)$$N_{i,min}=10^{\left(c_{i}\cdot {log}_{10}\left(N_{0}\right)-d_{i}\cdot H_{i}\right)}\left(1+b{\left(pH-pH_{opt}\right)}^{2}\right)\left(1+\frac{{log}_{10}(\eta )}{{{log}_{10}(\eta }_{max})}\right)$$

These equations describe the effects of the process parameters intensity (*H_i_*), pH, and viscosity (*η*) of the medium on the kinetic (*k_i,max_*) and equilibrium parameters under steady-state conditions (*N_i,min_*). Logistic and log-linear equations are used to describe how the inactivation rate changes with parameter *H_i_.* In Equation (4), the inactivation rate approaches zero when a certain critical intensity level (*h_i,c_*) is reached. If the intensity is increased above the critical level, *k_i_* increases at a rate *w_i_* until the value of *k_i,max_* is reached. The log-linear Equation (5), in which *c_i_* and *d_i_* represent classical coefficients that play a similar role to their counterparts *h_i,c_* and *w_i_*, describes how the residual microbial load varies with the *H_i_* of the treatment and the initial contamination (*N*_0_).

Figure 1 shows a schematic diagram of the processes, providing a simplified representation of the model using the System Dynamics language.

If the food is treated with the HPP technique, the intensity (*H_HPP_*) corresponds to the operating pressure applied to the system in MPa. For US treatment, the acoustic power, in Watts, was chosen as the process parameter (*H_US_*). The intensity factors are given in Equation (1) for US and Equation (6) for *HPP*:(6)$$H_{HPP}=P$$

The *H_US_* depends strongly on the type of ultrasound device and its maximum amplitude (*A_max_*), frequency, and configuration. The acoustic power can be easily determined by Equations (1) and (7), with a linear correlation between amplitude percentage (*A_perc_*) and (*d**T*/*d**t*).
(7)$$\frac{dT}{dt}=m\cdot \frac{A_{perc}\cdot A_{max}}{100}+q$$

To estimate *H_US_* by using Equations (1) and (7), a preliminary set of experiments was conducted as described in Section 2.2. The value of *m* and *q* were estimated equal to 0.1 °C s^−1^ and 20 °C (Appendix A).

In addition, other definitions might also be useful during the scale-up phase: intensity (W cm^−2^) and power density (W mL^−1^). Starting from the acoustic power (*H_US_*), the former is calculated by dividing by the surface of the sonotrode (cm^2^) and the latter by the volume of the sample (mL).

The second term of Equations (4) and (5) is useful to model the dependence of microbial growth on different pH values. For this reason, a quadratic empirical model was used where a represents the parameter that defines the size of the pH range in which a given microorganism can grow, assuming a certain optimal pH (pH_opt_) as the middle point of this range.

As the pH approaches extreme values (e.g., 0 or 14), k_i_ increases, while *N_i,min_* decreases. The last term of Equations (4) and (5) instead shows the dependence of the model on *η* (cP). If the viscosity value increases towards a maximum value *η_max_*, *k_i_* decreases, while *N_i,min_* tends to increase.

To evaluate the predictive ability of the model, it was necessary to compare the simulations with accurate experimental data, going through well-defined phases: an appropriate selection of the model’s structural coefficients, parametric estimation when operating conditions vary, and validation against a new set of experimental data.

#### 2.5.1. Structural Parameters

The parameters to be calibrated were selected by evaluating the nature and impact of each coefficient change on the model results. Table 1 lists the parameters for this sensitivity analysis.

The local sensitivity of the model was assessed by varying each parameter individually by ±10%. The sensitivity index was calculated using equation [27,28]:(8)$${SSE}_{m,\Delta }=\sqrt{\frac{1}{m}\sum_{i=1}^{N}\frac{{\left({C_{j}\left(p_{1},\ldots ,p_{m}+\Delta ,\ldots ,p_{n}\right)}_{.i}-{C_{j}(p_{1},\ldots ,p_{n})}_{.i}\right)}^{2}}{{C_{j}\left(p_{1},\ldots ,p_{n}\right)}_{,max}-{C_{j}\left(p_{1},\ldots ,p_{n}\right)}_{,min}}}$$

*SSE_m_*_,Δ_ represents the influence of a single parameter *p* on microbial concentration (*X*), *N* is the number of pieces of experimental data in a single test, and *m* is the number of parameters to be analyzed.

#### 2.5.2. Parametric Estimation

The sum of the squared errors (*SSE*) given in Equation (9) was used to calibrate the model and evaluate the discrepancy between the experimental and the simulated data:(9)$$SSE=\sum_{i=1}^{N}\frac{{\left(X_{data,i}-X_{sim,i}\right)}^{2}}{X_{data,max}}$$
where *X* represents the microbial load in CFU mL^−1^. The first parametric estimation was performed on food treatment data using the US technique, varying the power (i.e., amplitude in µm) applied to three different products (i.e., apple, orange, and carrot juice) previously contaminated with *Escherichia coli* (see Section 2.2 for details). The second parametric estimation was performed on a dataset describing the treatment of orange juice with the HPP technique at operating pressures of 300 MPa and 400 MPa (see Section 2.3 for details).

#### 2.5.3. Model Validation

The goodness of the parametric estimation was assessed by validating the model with a new set of experimental data using a combination of US and HPP techniques (see Section 2.4 for details).

### 2.6. Microbial Analysis

The treated juice samples were serially diluted, and the *E. coli* population was measured by pipetting 100 μL of samples onto agar media (HIMEDIA; product no. M091A) and incubating the plated dishes for 24 h at 37 °C. Upon incubation, the colonies were counted. The results are expressed as the colony-forming units per mL of juice (CFU mL^−1^).

### 2.7. Viscosity, pH, and Brix Measurement

The dynamic viscosity, pH, and Brix of the treated orange juice samples were measured to examine the effect of the ultrasound application, high-pressure processing, or their combination on product quality. Viscosity was measured using a dial viscometer (Brookfield, Middleboro, MA, USA), pH using a pH meter (METTLER TOLEDO, Barcelona, Spain), and total soluble solids were estimated as °Brix using a Refractometer (HANNA, Szeged, Hungary). Samples for pH and Brix were thawed, and measurements were made at 25 °C.

### 2.8. Data Analysis

All the microbial experiments were conducted in triplicate, and the data are presented as mean ± standard deviation. An ANOVA was performed, and the means were compared using Tukey’s test. The significance of the values was determined at a 5% level. Mathematical equations were integrated using MATLAB R2024a with a variable order solver (ode15s, [29]). The minimization was performed using the MATLAB routine fminsearch, which implements the Nelder–Mead simplex algorithm.

## 3. Results and Discussion

### 3.1. Acoustic Inactivation of Microorganisms

The obtained effect of ultrasound application on microorganisms depended on external and intrinsic factors and on the type and characteristics of the microorganism. It is important to consider the acid adaptation and pathogen strain, which influence ultrasound inactivation [30]. The preliminary results provided important information about the selection on the most indicated probe. For technical purposes, a good balance between acoustic intensity and acoustic power was pursued. Probe C, with a nominal amplitude (100%) of 164 um, delivered a low acoustic power (<80 W); however, the calculated acoustic intensity was very high (>200 W cm^−2^), which promoted the excessive heating of the sample. Therefore, only probe A and probe B, with nominal amplitudes (100%) of 46 µm and 99 µm, were used for the inactivation experiments. Table 2 collects the *E. coli* log reduction achieved when the inoculated juice was acoustically treated with probe A and probe B.

As expected, the ultrasound application promoted a reduction in the *E. coli* population from the first moment it was applied. The initial *E. coli* population in the juice was 5.1 log CFU mL^−1^. After 22.1 min (1329 s) of acoustic application, the population was reduced to 3.4 log CFU mL^−1^, being the log-red of 1.71 (probe A) and 1.67 (probe B). The microbial inactivation continued until 1.3 log CFU mL^−1^, and it was reached after 66.5 min. The microbial inactivation could be caused by the collapse of cavitation bubbles, the development of free radicals (e.g., OH^–^), and the production of hydrogen peroxide [31]. When comparing both probes, the inactivation effect was quite similar; this could be explained by the acoustic power delivered, which ranged between 192 and 196 W. The acoustic energy that reached the samples was transformed into heat, and heating was inevitable. Both samples experienced a similar temperature increment from 10 °C (initial) to 20 °C (22.2 min), and it was kept constant until the end of the experiment due to the use of the cooling system.

Valero et al. [32] studied the impact of US application (150, 300, and 600 W at 23 kHz or 120 and 240 at 500 kHz, for 15 min) and conventional heating on the inactivation of the endogenous contamination of orange juice. After ultrasound application, the mesophilic reduction was only 1.1 log CFU mL^−1^ and 1.7 log CFU mL^−1^, with an increment in temperature from 10 °C to 51 °C and 88 °C, respectively. Gómez-López et al. [33] studied US impact on orange juice at 10 °C from 2 to 10 min at a frequency of 20 kHz and acoustic amplitudes of 59.5, 71.4, and 89.2 μm. They reported that the mesophilic aerobic and yeast populations exhibited a reduction of 0.4 and 1.4 log CFU mL^−1^, respectively, during this short treatment (10 min), the highest reduction being observed when the juice was sonicated with the highest amplitude (89.2 μm). In general, when inoculated microorganisms are used, these show lower resistance than the natural flora of the juice. Salleh-Mack et al. [8] studied the effect of ultrasound pasteurization (24 kHz, 400 W) on the inactivation of *E. coli*. in a fruit juice model (pH 2.5, 12 °Brix). The initial *E. coli* population ranged from 6 to 8 log CFU mL^−1^, and the ultrasound pasteurization promoted a reduction of 5.6 (60.1 °C, 3 min) and 5.1 log CFU mL^−1^ (29.9 °C, 11 min). Ugarte-Romero et al. [31] applied power ultrasound (20 kHz, 46 µm) to apple cider for 5 min at temperatures between 40 and 60 °C and showed a 5.3 log reduction of *E. coli* at 40 °C, while at 60 °C the results were not different from the ones obtained with thermal treatment.

Considering the similar inactivation effects observed with probes A and B, a new set of experiments was proposed using only probe B, which had a higher amplitude (99 μm) and, due to the smaller probe area (14 mm), a higher acoustic density (Wcm^−2^) compared to probe A (22 mm). Figure 2 (left) collects the effect of the acoustic amplitude on *E. coli* microbial reduction when power ultrasound was applied using probe B at 50% (46 µm) and 100% (99 µm) amplitude. It can be seen in the Figure 2 that after 720 s of application, the microbial reduction was of 0.65 log and 1.34 log CFU mL^−1^ when the amplitude was 46 and 99 µm. After 60 min of sonication, the *E. coli* reduction achieved at 99 μm (3.59 log CFU mL^−1^) doubled the one observed at 46 μm (1.59 log CFUmL^−1^). As expected, the higher amplitudes resulted in the more effective creation of cavitation, therefore a higher microbial reduction. The ultrasonic effect was profound at higher ultrasonic amplitudes, which corresponded to a higher acoustic power, being of 98 and 195 W at 46 and 99 µm, respectively.

The data have been used to calibrate the model (Equations (4) and (5)). The result of the parametric estimation highlighted the need to separate the datasets with respect to a specific US usage power, resulting instead in being non-specific for the other product characteristics. The results in Figure 2 (right) show a perfect agreement between experimental (ln(XData)) and simulated data (ln(XSim)) (R^2^ = 0.98). The values of the coefficients obtained by the estimation are given in Table 3 for the different operational performances.

### 3.2. HPP Inactivation of Microorganisms

Surely, like for US treatment, the obtained lethal effect of HPP processing on microorganisms also depends on external and intrinsic factors, and, also, on the type and characteristics of the microorganism. The effect of different HPP treatments on *E. coli* inactivation in the inoculated orange juice is presented in Table 4, and as seen, the combination of pressure level and pressure time has a varying effect on survival. Generally, though, longer HPP processing results in a higher level of inactivation. Hence, increasing the pressure time from 1 min to 10 min at 300 MPa increases the log reduction from 0.16 ± 0.02 (mean of the three batches A, B, and C @ HPP time 1 min) to 4.0 ± 0.1 (mean of the three batches A, B, and C @ HPP time 10 min). The HPP level at 400 MPa needed a shorter time to reduce the *E. coli* population; thus, 2 min at 400 MPa produced a ~4-log inactivation (3.9 ± 0.8, mean of the three batches A, B, and C @ HPP time 2 min). The inoculum of *E. coli* was completely inactivated by a treatment of 400 MPa for 3 min. The different loss of viability at identical pressures and times for batches A, B, and C may be due to some temporal distribution of lethal effects. The pressure-induced killing mechanism is based on changes in intracellular components. Therefore, the proportion of the cells that are sub-lethally injured may vary to some extent at these short pressure treatments, and they may still be able to repair themselves until the analysis. On the other hand, HPP at 400 MPa for 3 min is enough to damage all cells leading to total death. The data have been used to calibrate the model (Equations (3) and (4)). In this case, it was not necessary to separate the data in terms of operating pressure to calibrate the coefficients; as shown in Figure 3, overall, a good agreement was obtained between the experimental data and the simulated ones (R^2^ = 0.88). The result of the parametric estimation is shown in Table 5.

The pressure-induced inactivation of *E. coli* in orange juice has been studied previously, and our results agree with those reported by Jordan et al. [34] and Bayindirli et al. [35]. Bayindirli et al. [35] found that after treatment at a low pressure of 250 MPa for 5, 10, and 20 min (at 30 °C) a survival of 3.18-log, 3.04-log, and 1.94-log, respectively, was still present in the juice. Upon increasing the pressure level to 350 MPa (5 min) and processing temperature to 40 °C, a complete inactivation of *E. coli* O157:H7 933 was obtained. Similarly, after pressure treatment at 350 MPa for 5 min, no survivors of the *E. coli* O157 strain NCFB 1989 were detected in orange juice [34]. However, strain C9490 was significantly more resistant to pressure than strain NCFB 1989, and even HPP treatment at 500 MPa for 5 min only resulted in a 1–2 log reduction in orange juice.

### 3.3. US + HPP Inactivation of Microorganisms

A combination of US 1440 s + HPP 400 MPa for 2 min showed a similar log reduction (5 log) compared to that of US 2160 s + HPP 400 MPa for 2 min. The fact that 1440 s (250 Ws mL^−1^) of US treatment promotes a similar or slightly higher microbial reduction than 2160 s (375 Ws mL^−1^) is very important in terms of the optimization of the combination of both technologies. The results of ultrasound application (US) following high-pressure processing (HPP) in the killing of *E. coli* is reported in Table 6. The US treatment is more effective than HPP at a low duration (1 min), while HPP for longer time (5 and 10 min) is more efficient than US processing. Interestingly, the combined US + HPP treatment seems to act synergistically and results in no detectable colony-forming units per mL of juice, indicating a very high inactivation of the *E. coli* bacteria. The U.S. Food and Drug Administration (FDA) typically recommends achieving a minimum of a 5-log (i.e., 99.999%) inactivation of *E. coli* O157 for fruit juices, including orange juice. This level of inactivation is intended to ensure safety from pathogenic microorganisms.

The validation simulation was performed by first simulating the microbial load with the US treatment for and then giving the initial microbial population value in the HPP treatment as the final US treatment value. The validation results are shown in Figure 4, which shows the comparison of the natural logarithm of the simulated microbial concentration, obtained from the calibration coefficients of the two techniques, and the natural logarithm of the experimental data in the parity plot. As can be seen, the results show a good agreement, with a R^2^ equal to 0.82. In general, as the US operating time increases, the treatment tends to be overestimated by the model, while the HPP is slightly underestimated.

### 3.4. Viscosity, pH, and Brix

The soluble solids content and pH of the control orange juices were 10.70–12.43 Brix and 3.55–3.61, respectively. As expected, neither the HPP nor US + HPP treatments affected these small compounds; hence, no marked differences were observed for soluble solids and pH in all the treated and untreated orange juices. This finding agrees with previous investigations that combined US and HPP treatment of juices such as apple [7], cranberry [36], and carrot [37]. The juice’s rheological properties are directly related to structure, particle size, and composition and the interaction between them. Acoustically treated samples exhibited a reduction in dynamic viscosity from 23.1 cP (untreated sample) down to 6.3–4.7 cP. In general, the higher the acoustic energy applied, the lower the viscosity of the sample. However, the effect of US on the sample viscosity can be temporary or permanent [38].

### 3.5. Selection of the Model’s Structural Coefficients

Figure 5 shows the result of the influence of a single parameter p on microbial concentration (Equation (8)). The parameters a, b, and ηmax are all related to the system response as a function of pH (a and b) and sample viscosity. Due to the lack of experimental evidence to delve into the variability in the results at different pH ranges and viscosities, the above parameters were excluded from the parameter estimation. On the other hand, variation in parameter n, the index associated with the resistant subpopulation *N*_i,min_, has such a significant effect on US that it dominates and crucially minimizes the effects of all other parameters. For these reasons, the above coefficients were excluded from the calibration. Table 7 displays the fixed parameter values of the model.

For US and HPP treatments, the variation in the remaining parameters shows that the global influence on the model can only be achieved by changing kmax, hc, and d.

## 4. Conclusions

This study investigated the effect of combining ultrasound (US) and high-pressure processing (HPP) on the inactivation of *Escherichia coli* in orange juice. Starting from an initial population of 5 log CFU mL^−1^, US alone reduced the count to 1 log CFU mL^−1^ after 66 min, while the juice temperature increased from 10 to 22 °C due to acoustic energy. Acoustic amplitude and intensity were critical for microbial inactivation. HPP at 300 MPa showed that extending the treatment time from 1 to 10 min increased the log reduction from 0.16 to 4.03 CFU mL^−1^. At 400 MPa, a shorter treatment of 2 min achieved an approximately 4-log inactivation, with complete inactivation at 3 min. Combining US (1440 s) and HPP (400 MPa for 2 min) resulted in a 5 log CFU mL^−1^ reduction, indicating a synergistic effect.

The US and HPP processes have been mathematically modeled to simulate the inactivation curves of target microorganisms as functions of processing time and intensity. The model has been calibrated and validated for orange juice and can be extended to different microorganisms and enzymes. It accounts for both the downward and upward concave behaviors typical of sigmoid curves, allowing for the prediction of trends based on model parameters. This modeling study serves as a foundation for predicting microbial load reduction and optimizing process parameters for specific targets.

## Figures and Tables

**Figure 1 foods-13-03463-f001:**
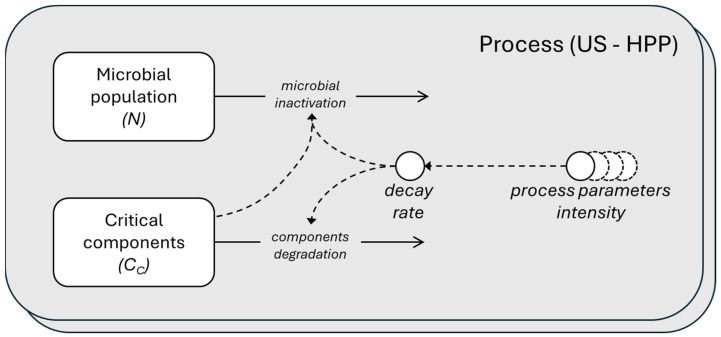
Schematic representation in System Dynamics language of the microbial population and critical components dynamics. The microbial inactivation and components degradation depending on the decay rate that is function of the parameter intensity of the two processes, US and HPP. The rectangles are the stocks (the ODE state variable), the circles are the parameters, and the solid-lined arrows represent the flows, while the dashed-lined arrows represent the influence.

**Figure 2 foods-13-03463-f002:**
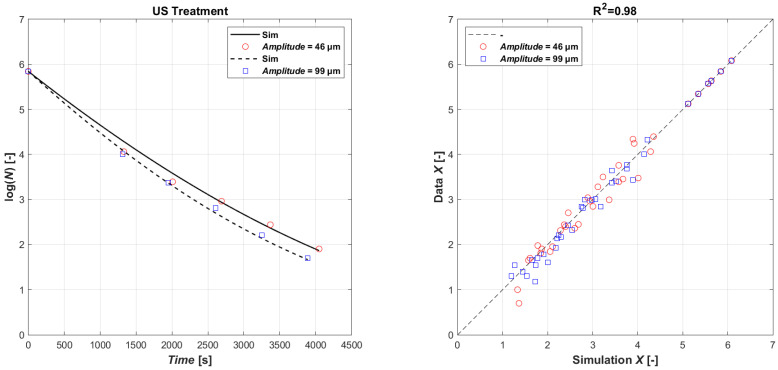
Model fitting of experimental data (**left**): time series of measured *E. coli* microbial mass with power ultrasound with amplitude 46 µm (open red circle symbol) and 99 µm (open blue square symbol) data vs. model simulations (continuous lines and dashed lines, respectively); parity plot US (**right**).

**Figure 3 foods-13-03463-f003:**
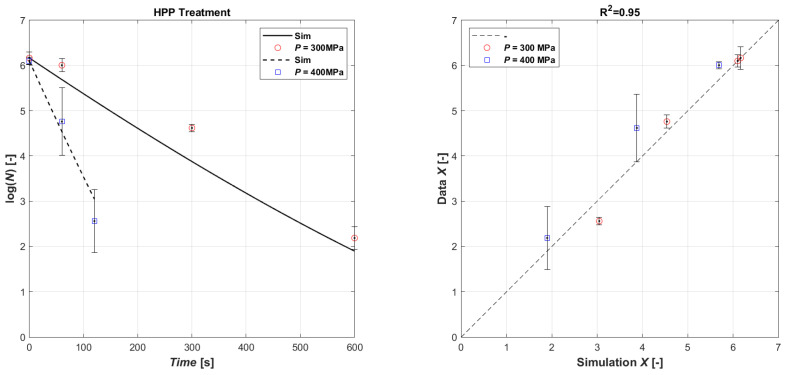
Model fitting of experimental data (**left**): time series of measured *E. coli* microbial mass with HPP 300 MPa (open red circle symbol) and with HPP 400 MPa (open blue square symbol) data vs. model simulations (continuous lines and dashed lines, respectively). Parity plot HPP (**right**).

**Figure 4 foods-13-03463-f004:**
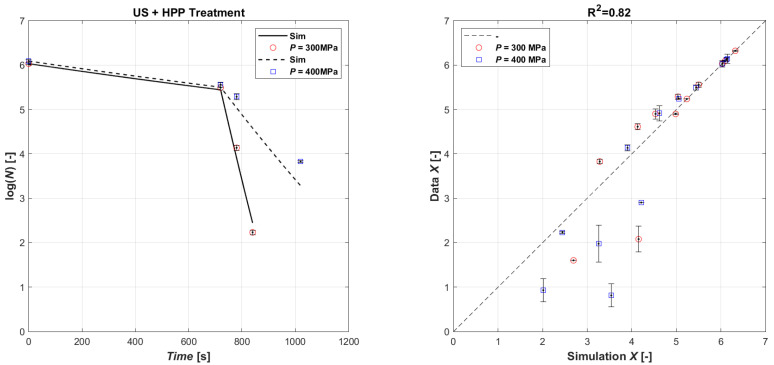
Model validation of experimental data (**left**): time series of measured *E. coli* microbial mass with HPP 400 MPa (open red circle symbol) and HPP 300 MPa (open blue square symbol) data vs. model simulations (continuous lines and dashed lines, respectively); parity plot US (720 s) + HPP (**right**).

**Figure 5 foods-13-03463-f005:**
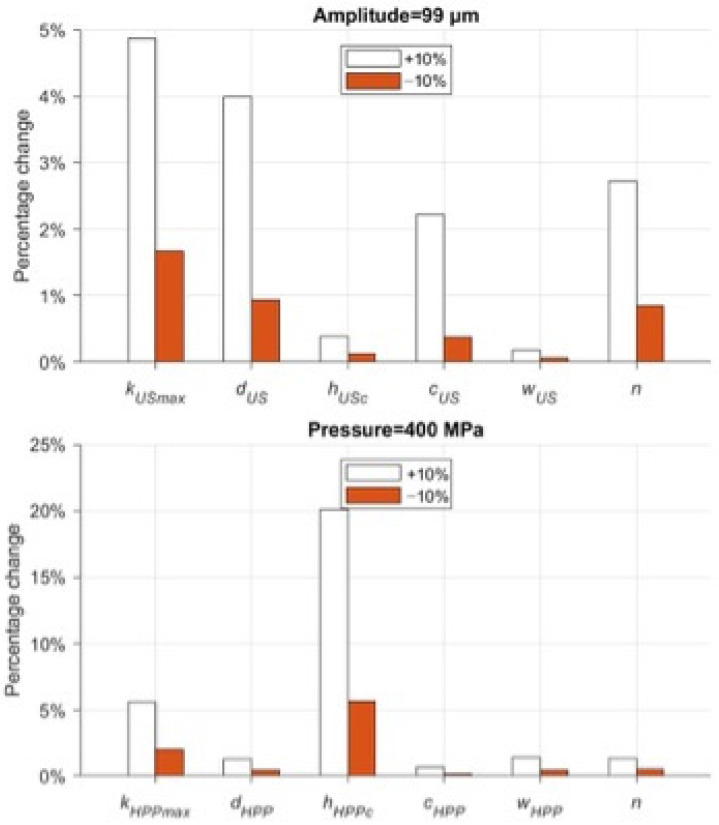
Local sensitivity analysis of model’s parameters in US and HPP treatments.

**Table 1 foods-13-03463-t001:** Model parameters.

Parameter	Value	Units
$k_{US,max}$	10	$s^{-1}$
$h_{US,c}$	250	W
$w_{US}$	$1\cdot 10^{-4}$	$W^{-1}$
$c_{US}$	1	−
$d_{US}$	$3\cdot 10^{-2}$	$W^{-1}$
$k_{HPP,max}$	15	$s^{-1}$
$h_{HPP,c}$	$4\cdot 10^{2}$	MPa
$w_{HPP}$	$1.5\cdot 10^{-2}$	${MPa}^{-1}$
$c_{HPP}$	0.7	−
$d_{HPP}$	$1.1\cdot 10^{-2}$	${MPa}^{-1}$
$\eta_{max}$	$1\cdot 10^{5}$	cP
m	0.1	−
a	$1\cdot 10^{-2}$	−
b	$1\cdot 10^{-4}$	−

**Table 2 foods-13-03463-t002:** *E. coli* abundance and log reduction of inoculated orange juice as function of US treatment time and sonotrode.

Probe	Time	Temperature	*E. coli*	Reduction
	[s]	[°C]	[log CFU mL^−1^]	[log CFU mL^−1^]
Probe A (46 μm)	0	10	5.1	0
	1329	20	3.4	1.71
	1989	20	3.0	2.12
	2649	20	2.1	2.99
	3329	20	1.6	3.47
	3988	20	1.3	3.82
Probe B (99 μm)	0	10	5.1	0
	1329	20	3.4	1.67
	2009	20	2.8	2.28
	2689	20	2.4	2.72
	3369	20	1.8	3.31
	4044	20	1.7	4.42

**Table 3 foods-13-03463-t003:** Calibrated US parameters.

Parameter	Value (46 µm)	Value (99 µm)
$k_{US,max}$	$8.90\cdot 10^{-3}$	$1.10\cdot 10^{-2}$
$h_{US,c}$	392.95	755.83
$d_{US}$	0.18	0.10

**Table 4 foods-13-03463-t004:** *E. coli* abundance and log reduction of inoculated orange juice as function of HPP pressure level and treatment time (*n* = 6).

Pressure [MPa]	Time [min]	*E. coli* [log CFU mL^−1^]	Reduction [log CFU/mL]
300	0 (control)	6.2 ± 0.6	
	1	6.0 ± 0.5	0.16 ± 0.02
	5	4.6 ± 0.4	1.5 ± 0.1
	10	2.2 ± 0.2	4.0 ± 0.1
400	0 (control)	6.2 ± 0.5	
	1	4.7 ± 0.5	1.9 ± 0.8
	2	2.5 ± 0.2	3.9 ± 0.8
	3	ND *	-

* ND: no colonies detected.

**Table 5 foods-13-03463-t005:** Calibrated HPP parameters.

Parameter	Value
$k_{HPP,max}$	$2.10\cdot 10^{-2}$
$h_{HPP,c}$	427.86
$d_{HPP}$	0.23

**Table 6 foods-13-03463-t006:** *E. coli* abundance and log reduction of inoculated orange juice as function of combined US and HPP treatment (*n* = 2).

Pressure (MPa)	Time (min)	US (s)	*E. coli* (log CFU mL^−1^)	Reduction (log CFU mL^−1^)
300	0	720	5.5 ± 0.4	0.54 ± 0.05
		1440	5.2 ± 0	1.09 ± 0
		2160	5.2 ± 0.3	1.14 ± 0.06
	1	720	5.3 ± 0.3	0.81 ± 0.05
		1440	4.9 ± 0.4	1.42 ± 0.01
		2160	3.6 ± 0.4	1.42 ± 0.01
	5	720	3.8 ± 0.3	2.27 ± 0.04
		1440	2.0 ± 0.1	4.33 ± 0.4
		2160	1.6 ± 0.1	4.42 ± 0.1
	10	720	ND *	
		1440	ND *	
		2160	ND *	
400	0	720	5.5 ± 0.5	0.54 ± 0.03
		1440	5.2 ± 0.1	0.89 ± 0.13
		2160	4.9 ± 0.3	1.21 ± 0.07
	1	720	4.1 ± 0.3	1.90 ± 0.05
		1440	1.0 ± 0.1	3.2 ± 0.1
		2160	1.0 ± 0.1	4.1 ± 0.1
	5	720	2.0 ± 0.1	3.80 ± 0.02
		1440	1.0 ± 0.1	5.4 ± 0.5
		2160	1.0 ± 0.1	5.20 ± 0.06
	10	720	ND *	
		1440	ND *	
		2160	ND *	

* ND: no colonies detected.

**Table 7 foods-13-03463-t007:** Fixed parameters.

Parameter	Value
$\eta_{max}$	$1.0\cdot 10^{5}$
m	0.10
$a_{1}$	$1.0\cdot 10^{-2}$
$a_{2}$	$1.0\cdot 10^{-4}$

## Data Availability

The original contributions presented in this study are included in the article/Appendix A. Further inquiries can be directed to the corresponding author.

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
