# Peer review of "A Mathematical Model for the Combination of Power Ultrasound and High-Pressure Processing in the Inactivation of Inoculated E. coli in Orange Juice"

_foods, 2024, doi:10.3390/foods13213463_

Round 1
Reviewer 1 Report
Comments and Suggestions for Authors
Line 111: E. coli and other bacterial name should be italic。Please check the whole manuscript, as well as figures and tables.
Line 331: dot is missing after juice.
Line 361: Is it Figure 2B?
Table 2: why the value of table 2 doesn’t have the standard deviation like table 4.
The resolution of Figure 3 is very low.
The content of Table 6 is misplaced.
Figure 4 (left) is difficult to understand. I couldn’t connect the data of table 6 and Figure 4 (left). Is the symbol of 300 MPa and 400 MPa exchanged.
Increase the discussion of the inactivate effect and the mathematical model for the combination of US and HPP treatment.
Comments on the Quality of English Language
The English usage is good, but not quite up to the level required for publication. If the Journal does not edit to correct , it is incumbent on the authors to do so.
Author Response
Response to Reviewer 1 Comments |
||
1. Summary |
|
|
Thank you very much for taking the time to review this manuscript. Please find the detailed responses below and the corresponding revisions/corrections highlighted/in track changes in the re-submitted files. |
||
2. Questions for General Evaluation |
Reviewer’s Evaluation |
Response and Revisions |
Does the introduction provide sufficient background and include all relevant references? |
Can be improved |
The manuscript has been revised to improve introduction, results and conclusion |
Is the research design appropriate? |
Can be improved |
|
Are the methods adequately described? |
Can be improved |
|
Are the results clearly presented? |
Can be improved |
|
Are the conclusions supported by the results? |
Can be improved |
|
3. Point-by-point response to Comments and Suggestions for Authors |
||
Comments 1: Line 111: E. coli and other bacterial name should be italic。Please check the whole manuscript, as well as figures and tables. |
||
Response 1: Thank you for pointing this out. We agree with this comment. Therefore, we have checked, and all the microbial stains are now written in Italia. |
||
Comments 2: Line 331: dot is missing after juice. |
||
Response 2: Thank you for pointing this out. The manuscript has been emended as reviewer suggested. Comments 3: Line 361: Is it Figure 2B? |
||
Response 3: Thank you for pointing this out. Yes, it was an error and now the manuscript has been emended. Comments 4: Table 2: why the value of table 2 doesn’t have the standard deviation like table 4 Response 4: Thank you for pointing this out. Actually 2 replications have been performed. Comments 5: The resolution of Figure 3 is very low. Response 5: I have uploaded a new picture Comments 6: Table 2: The content of Table 6 is misplaced. Response 6: The table has been replaced and emended Comments 7: Figure 4 (left) is difficult to understand. I couldn’t connect the data of table 6 and Figure 4 (left). Is the symbol of 300 MPa and 400 MPa exchanged. Response 7: Thank you for pointing this out. It was emended. Now should be clear. Comments 8: Increase the discussion of the inactivate effect and the mathematical model for the combination of US and HPP treatment. Response 8: Thank you for the suggestion, the discussion has been improved. |
Reviewer 2 Report
Comments and Suggestions for Authors
The effect of ultrasound application combined with high pressure processing on the inactivation of Escherichia coli inoculated in orange juice was studied by a mathematical model. The topic of this work is interesting. Some modifications are recommended.
Microbial strains throughout the manuscript should be written in italic.
HPP in line 42 should be defined. Lines 42-46: The stabilization mechanisms of HPP are not clearly introduced. Meanwhile, the factors affecting the stabilization effect of HPP are not stated.
Lines 60-70: The examples are unnecessary. The authors should summarize the importance of combinedly using HPP and US.
Lines 71-73: This section is too simple and should be extended.
Line 74: It has been shown... Check the language throughout the manuscript.
The methods are described well.
Table 2: Probe A and Probe B are not defined. The same for Probe C in the text.
Some figures are not very clear, such as Figs. 2-4.
At last, the authors should provide the suitable mathematical model.
The conclusion should be shortened.
Comments on the Quality of English Language
The English could be improved to more clearly express the research.
Author Response
1. Summary |
|
Thank you very much for taking the time to review this manuscript. Please find the detailed responses below and the corresponding revisions/corrections highlighted/in track changes in the re-submitted files. |
|
2. Questions for General Evaluation |
|
Does the introduction provide sufficient background and include all relevant references? |
|
Is the research design appropriate? |
|
Are the methods adequately described? |
|
Are the results clearly presented? |
|
Are the conclusions supported by the results? |
|
The effect of ultrasound application combined with high pressure processing on the inactivation of Escherichia coli inoculated in orange juice was studied by a mathematical model. The topic of this work is interesting. Some modifications are recommended. 3. Point-by-point response to Comments and Suggestions for Authors |
|
Comments 1: Microbial strains throughout the manuscript should be written in italic. |
|
Response 1: Thank you for pointing this out. We agree with this comment. Therefore, We have checked, and all the microbial stains are now written in Italia. |
|
Comments 2: HPP in line 42 should be defined. Lines 42-46: The stabilization mechanisms of HPP are not clearly introduced. Meanwhile, the factors affecting the stabilization effect of HPP are not stated. |
|
Response 2: Thank you for pointing this out. The introduction has been emended as reviewer suggested: L44-48: The main stabilization mechanisms involved in HPP are cell mebrane distruction, protein denaturation, DNA damage. The main factors affecting stabilization effect of HPP are the pressure level, duration of treatment, temperature, pH level, food composition, microbial resistance and food structure. Comments 3: Lines 60-70: The examples are unnecessary. The authors should summarize the importance of combinedly using HPP and US. |
|
Response 3: Thank you for pointing this out. The manuscript has been emended as suggested: L64-69 The combined use of High-Pressure Processing (HPP) and ultrasound (US) significantly enhances microbial inactivation in food products. Increasing the amplitude of ultrasonic waves and applying higher pressure can improve the effectiveness of US [14]. When used together, HPP and US can exhibit both additive and synergistic effects, leading to greater reductions in pathogens [7, 15, 16,17]. This integration enhances food safety and preservation, making it a powerful approach in food processing. Comments 4: Lines 71-73: This section is too simple and should be extended. Response 4: Thank you for pointing this out. The manuscript has been emended as suggested: L70-83 Mathematical modeling of the HPP and ultrasound processes is an area of significant interest, as it enables the detailed analysis and understanding of the dynamic behavior of various species within food products, including microorganisms, enzymes, and other bioactive compounds. These models are crucial for process control, optimization and prediction. In fact, mathematical models provide a framework for monitoring and controlling the conditions of HPP and US treatments. By simulating different scenarios, operators can identify optimal conditions that maximize microbial inactivation while preserving the quality of the food. Moreover, through modeling, it is possible to optimize parameters such as pressure, temperature, treatment time, and ultrasonic amplitude. This ensures that the processes are both effective and efficient, minimizing energy usage and maximizing product quality. Models can also predict the outcomes of various processing conditions on microbial inactivation and the stability of food components. This predictive capability is essential for designing processes that meet safety regulations and consumer expectations Comments 5: Line 74: It has been shown... Check the language throughout the manuscript. Response 5: Thank you for pointing this out. The manuscript has been emended as suggested: L84-86: It has been demonstrated multiple times that when microorganisms are exposed to new nonthermal technologies, inactivation does not follow first-order kinetics. In this context, classical deterministic models based on first order kinetics are no longer applicable Comments 6: The methods are described well. Response 6: Thank you for pointing this out. Comments 7: Table 2: Probe A and Probe B are not defined. The same for Probe C in the text. Response 7: Thank you for pointing this out. The probes are defined in the text at L 128-129. Details were also included in table 2. Comments 8: Some figures are not very clear, such as Figs. 2-4. Response 8: Thank you for pointing this out. I think the problem in this case depends on the submission process, in any case I will provide new images in case higher quality figures are needed. Comments 9: At last, the authors should provide the suitable mathematical model. Response 9: We thank the reviewer for the comment. The values of fixed the parameters set in the model, described in Section 3.5, have been included in the text in tabular form to provide all the necessary details of the mathematical model. Comments 10: The conclusion should be shortened. Response 10: Thank you for pointing this out. The conclusion has been shortened |
Reviewer 3 Report
Comments and Suggestions for Authors
The authors developed a mathematical model for inactivation of E. coli based on the experiment. It provided valuable information for industry.
However, the authors should give major revision based on the comments as followed,
1. Title, "non-thermal technologies", can the authors use the specific terms? Like "ultrasonic or/and high pressure processing" .
2. line 24, "2" should be superscripted. line 29, what is the abbrievation of ODE?
3. Species of bacterial should be italic, such as line 60, double check them through the article.
4. typeface should be consistent, line 296, in the table.
5. Table 6, it should be improved. "ND" should be noted or explained.
6. line 505-508 can be deleted.
7. reference list, some information are missing, such as pages or article code. Line 527, 532, and so on. The journal name can be abbreviated uniformly or not? Check line 544, 541, etc. Please check them carefully, and follow the requirements of journal strictly.
Author Response
1. Summary |
|
|
Thank you very much for taking the time to review this manuscript. Please find the detailed responses below and the corresponding revisions/corrections highlighted/in track changes in the re-submitted files. |
||
2. Questions for General Evaluation |
Reviewer’s Evaluation |
Response and Revisions |
Does the introduction provide sufficient background and include all relevant references? |
Can be improved |
The manuscript has been revised as referee suggestion to improve all the section |
Are all the cited references relevant to the research? |
Can be improved |
|
Is the research design appropriate? |
Can be improved |
|
Are the methods adequately described? |
Can be improved |
|
Are the results clearly presented? |
Can be improved |
|
Are the conclusions supported by the results? |
Can be improved |
|
3. Point-by-point response to Comments and Suggestions for Authors |
||
Comments 1: Title, "non-thermal technologies", can the authors use the specific terms? Like "ultrasonic or/and high pressure processing". |
||
Response 1: Thank you for pointing this out. The title of the manuscript has been emended as suggested. The new title proposed is: A mathematical model for the combination of power ultrasound and high-pressure processing in the inactivation of inoculated E. coli in orange juice |
||
Comments 2: line 24, "2" should be superscripted. line 29, what is the abbrievation of ODE? |
||
Response 3: Thank you for pointing this out. We have checked, and all the microbial stains are now written in Italia. Comments 4: typeface should be consistent, line 296, in the table. Response 4: Thank you for pointing this out. We have check it, and the manuscript was emended when required Comments 5: Table 6, it should be improved. Response 5: Thank you for pointing this out. The table has been emended as suggested. Comments 6: line 505-508 can be deleted. Response 6: Thank you for pointing this out. The lines were deleted Comments 7: reference list, some information are missing, such as pages or article code. Line 527, 532, and so on. The journal name can be abbreviated uniformly or not? Check line 544, 541, etc. Please check them carefully and follow the requirements of journal strictly. |
||
Response 7: Thank you for pointing this out. The reference list has been emended as suggested. |
Reviewer 4 Report
Comments and Suggestions for Authors
Please see the attachment.

Author Response
1. Summary |
|
Thank you very much for taking the time to review this manuscript. Please find the detailed responses below and the corresponding revisions/corrections highlighted/in track changes in the re-submitted files. |
|
2. Questions for General Evaluation |
|
Does the introduction provide sufficient background and include all relevant references? |
|
Is the research design appropriate? |
|
Are the methods adequately described? |
|
Are the results clearly presented? |
|
Are the conclusions supported by the results? |
|
3. Point-by-point response to Comments and Suggestions for Authors I have reviewed the manuscript foods-3262698 by Óscar Rodríguez and co-authors, “A mathematical model for the combination of non-thermal technologies in the inactivation of inoculated E. coli in orange juice.” The goal of the study was to examine the inactivation kinetics of Escherichia coli inoculated in orange juice in relation to power ultrasound (US), high-pressure processing (HPP), and the combination of these non-thermal technologies. A mathematical model was designed and implemented to predict these effects. The manuscript is interesting and contributes new knowledge regarding the inactivation kinetics of Escherichia coli inoculated in orange juice by ultra sound, high-pressure processing and combination of those processes. A few comments to improve the manuscript can be found below. |
|
Comments 1: The abstract could be improved. The authors could add some numerical findings, the strain of E.coli, and process parameters to strengthen it. |
|
Response 1: [Type your response here and mark your revisions in red] Thank you for pointing this out. The numerical finding is included, and the stain was specified. |
|
Comments 2: Keywords: Instead of the abbreviation ODE the full name should be used. |
|
Response 2: Thank you for pointing this out. It was emended Comments 3: The introduction provides a good background on the subject. The objectives of the study are written clearly. |
|
Response 3: Thank you for pointing this out. Comments 4: 2.1. Bacteria stock culture and inoculum: Comments 4a: Line 107. Please provide the collection where the strain of E. coli was acquired. Response 4a: Sorry for this oversight, the provider has been added. Comments 4b: It isn’t clear whether the juice was filtered or contained solid parts. Please clarify. Response 4b: The juice did not contain solid particles. Comments 4c: The authors should provide the pH value or titratable acidity of the juice, as well as Brix or conductivity. Response 4c: The pH and Brix are given as results in section 3.4. Comments 4d: Why did the authors choose the specific inoculum concentration? Was it based in some previously published source? Please elaborate. Response 4d: It was because the treatment should be assuring at least 5 LOG reduction Comments 4e: Is there a formal recommendation regarding the number of E.coli log10 cycles that should be inactivated by the applied process in orange juice? Response 4e: Yes, there are formal recommendations regarding the inactivation of E. coli in orange juice. The U.S. Food and Drug Administration (FDA) typically recommends achieving a minimum of 5-log (i.e., 99.999%) inactivation of E. coli O157 for fruit juices, including orange juice. This level of inactivation is intended to ensure safety from pathogenic microorganisms. Different regulatory agencies might have varying recommendations, so it's always good to check the specific guidelines relevant to your region or the specific process being used. It was included in the text to discuss the results (L 444-447) Comments 4f: Conclusion: The conclusions are consistent with the evidence and arguments presented. Response 4f: Thank you for pointing this out. Comments 5: Lines 488-489. The authors conclude that “The combined US+HPP treatment seems to act synergistically and results in no detectable colony-forming units per mL of juice” What was the detection limit of the survival count method? The authors should indicate it also in the diagrams. Response 5: ND (not detected) means 0 CFU mL-1. There were no colonies detected on the plate. The following formula for the calculation has been used: CFU/ml = (Number of colonies*dilution factor) / sample volume. Comments 6: Have the tested doses an effect on the acceptability of the juice? Response 6: Thank you for pointing this out. In this work, the acceptability of the juice was not studied. However, the activity was performed, and the results will be published in a following paper. Comments 7: The references are not appropriately cited, from what I could assess. Please modify according to the mdpi’s instructions. Response 7: Thank you for pointing this out. The reference has been modified according to the MDPI’s instructions. |
Round 2
Reviewer 1 Report
Comments and Suggestions for Authors
Line510-513: HPP at 300 MPa showed that extending the treatment time from 1 to 10 minutes increased log reduction from 0.16 to 4.03 CFU mL⁻¹. At 400 MPa, a shorter treatment of 2 minutes achieved approximately 4-log inactivation, with complete inactivation at 3 minutes.
The conclusion is not consist to Table 6 and Figure 4. The results of open red circle symbol should be 400 MPa. The time in Table 6 for 300 MPa should be 0, 1, 5, and 10 min.
Please check all the results carefully.
Author Response
Line510-513: HPP at 300 MPa showed that extending the treatment time from 1 to 10 minutes increased log reduction from 0.16 to 4.03 CFU mL⁻¹. At 400 MPa, a shorter treatment of 2 minutes achieved approximately 4-log inactivation, with complete inactivation at 3 minutes.
The conclusion is not consist to Table 6 and Figure 4. The results of open red circle symbol should be 400 MPa. The time in Table 6 for 300 MPa should be 0, 1, 5, and 10 min.
Please check all the results carefully.
Answer: I would like to thanks the reviewer for the comment. I have checked the results and amended the table 6 and the figure 4 as suggestion. it was an error reporting the plot and the tables.
Reviewer 3 Report
Comments and Suggestions for Authors
The authors have made substantial revision carefully based on the comments. Now it reads better. However, minor revision should be given before the publication.
1. In table, can the E. coli be exhibited with log CFU/mL not CFU/mL including the methods and results in article reported? CFU/mL or CFU mL-1 should be unified in Table 4 and 2, only one can be kept through the article, please check and correct.
2. Reference list section, the revision is not satisfactory, the missing or mistakes can be found. Line 543, 563, 571, missing page information, line 586, Vacciniumcorymbosum should be italic, and volume and page information missing, line 602, vol and pp should be deleted. Anyway, please double check and unify.
Author Response
Comment: In table, can the E. coli be exhibited with log CFU/mL not CFU/mL including the methods and results in article reported? CFU/mL or CFU mL-1 should be unified in Table 4 and 2, only one can be kept through the article, please check and correct.
Answer: All the results were reported in LOG CFU mL-1 and the unit was unified through the text .
Comment: Reference list section, the revision is not satisfactory, the missing or mistakes can be found. Line 543, 563, 571, missing page information, line 586, Vacciniumcorymbosum should be italic, and volume and page information missing, line 602, vol and pp should be deleted. Anyway, please double check and unify.
Answer: thanks to the reviewer for pointing out the errors. The manuscript was emended as referee suggested